# ARTS: Semi-Analytical Regressor using Disentangled Skeletal Representations for Human Mesh Recovery from Videos

Tao Tang
State Key Laboratory of General
Artificial Intelligence, Peking
University, Shenzhen Graduate School
Shenzhen, China
taotang@stu.pku.edu.cn

Hong Liu*
State Key Laboratory of General
Artificial Intelligence, Peking
University, Shenzhen Graduate School
Shenzhen, China
hongliu@pku.edu.cn

Yingxuan You
State Key Laboratory of General
Artificial Intelligence, Peking
University, Shenzhen Graduate School
Shenzhen, China
youyx@stu.pku.edu.cn

Ti Wang
State Key Laboratory of General
Artificial Intelligence, Peking
University, Shenzhen Graduate School
Shenzhen, China
tiwang@stu.pku.edu.cn

Wenhao Li
State Key Laboratory of General
Artificial Intelligence, Peking
University, Shenzhen Graduate School
Shenzhen, China
wenhaoli@pku.edu.cn

## ABSTRACT

Although existing video-based 3D human mesh recovery methods have made significant progress, simultaneously estimating human pose and shape from low-resolution image features limits their performance. These image features lack sufficient spatial information about the human body and contain various noises (e.g., background, lighting, and clothing), which often results in inaccurate pose and inconsistent motion. Inspired by the rapid advance in human pose estimation, we discover that compared to image features, skeletons inherently contain accurate human pose and motion. Therefore, we propose a novel semi-**A**nalytical **R**egressor using disen**T**angled **S**keletal representations for human mesh recovery from videos, called **ARTS**. Specifically, a skeleton estimation and disentanglement module is proposed to estimate the 3D skeletons from a video and decouple them into disentangled skeletal representations (i.e., joint position, bone length, and human motion). Then, to fully utilize these representations, we introduce a semi-analytical regressor to estimate the parameters of the human mesh model. The regressor consists of three modules: Temporal Inverse Kinematics (TIK), Bone-guided Shape Fitting (BSF), and Motion-Centric Refinement (MCR). TIK utilizes joint position to estimate initial pose parameters and BSF leverages bone length to regress bone-aligned shape parameters. Finally, MCR combines human motion representation with image features to refine the initial human model parameters. Extensive experiments demonstrate that our ARTS surpasses existing state-of-the-art video-based methods in both per-frame accuracy and temporal consistency on popular benchmarks: 3DPW, MPI-INF-3DHP, and Human3.6M. Code is available at https://github.com/TangTao-PKU/ARTS.

## CCS CONCEPTS

• **Computing methodologies** → **Reconstruction**.

## KEYWORDS

Human Mesh Recovery, Human Pose Estimation, Disentangled Skeletal Representations

**ACM Reference Format:**
Tao Tang, Hong Liu, Yingxuan You, Ti Wang, and Wenhao Li. 2024. ARTS: Semi-Analytical Regressor using Disentangled Skeletal Representations for Human Mesh Recovery from Videos. In *Proceedings of the 32nd ACM International Conference on Multimedia (MM '24), October 28-November 1, 2024, Melbourne, VIC, Australia.* ACM, New York, NY, USA, 10 pages. https://doi.org/10.1145/3664647.3680881

## 1 INTRODUCTION

Recovering human meshes from monocular images is a crucial yet challenging task in computer vision, with extensive applications in virtual reality, animation, gaming, and robotics. Different from 3D Human Pose Estimation (HPE) [1] that predicts the location of several skeleton joints, 3D Human Mesh Recovery (HMR) [2] is a more complex task, which aims to estimate the detailed 3D human mesh coordinates.

Although there has been some progress in 3D HMR from a single image [3–7], accurate and temporally consistent recovery of the human mesh from monocular videos remains challenging. With the success of the SMPL [8], previous video-based HMR methods directly predict SMPL parameters from image features. The SMPL model is a parametric human model consisting of 72 pose parameters and 10 shape parameters, which control 6890 vertices to form a human mesh. As shown in Figure 1 (a), previous video-based HMR methods first use the pre-trained Convolutional Neural Networks (CNNs) backbone [9] to extract image features from video frames, then design temporal networks based on Recurrent Neural

*Corresponding Author is Hong Liu (e-mail: hongliu@pku.edu.cn).

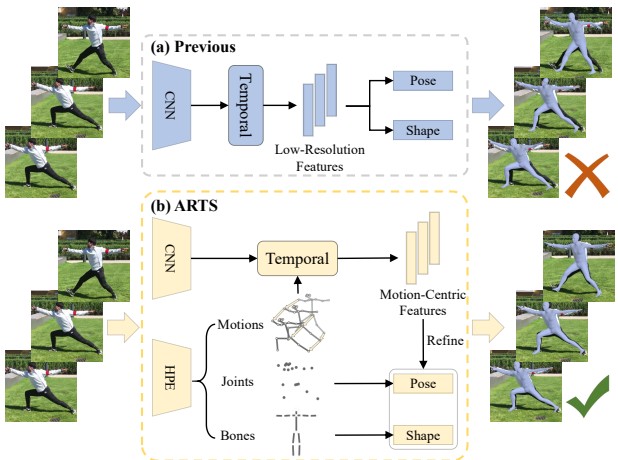

**Figure 1: Comparison between the previous video-based HMR methods and our ARTS. (a) Previous video-based HMR methods estimate the human pose and shape from low-resolution image features. (b) Our ARTS effectively utilizes disentangled skeletal representations (i.e., Motions, Joints, Bones) with image features to estimate and refine the human pose and shape.**

Networks (RNNs) [10–12] or Transformers [13–17] to extract the spatial-temporal coupled features. Finally, SMPL pose and shape parameters are obtained from the same low-resolution and coupled features using the fully connected layers. However, limited by the insufficient spatial information and noises in the image features, these methods tend to suffer from the following three problems:

**(i) Inaccurate pose estimation.** The image features extracted by ResNet [9] after global pooling are low-resolution with significant loss of spatial information, making it difficult for the following network to learn the highly non-linear mapping from image features to SMPL pose parameters [18–20].

**(ii) Ineffective shape fitting.** Many datasets [21–23] only contain a small number of subjects, resulting in the scarcity of body shapes. Consequently, employing neural networks to directly regress SMPL shape parameters often leads to overfitting [24], which often regresses average human shape during inference.

**(iii) Inconsistent human motion.** The image features contain various noises (e.g., background, lighting, and clothing) that affect the human motion capturing. Meanwhile, the changes in image features cannot directly reflect human movements, leading to undesirable motion jitters.

Inspired by the rapid advance in video-based HPE [25–29], we discover that the skeletons estimated by HPE algorithms can be utilized to alleviate the problems above. Compared to low-resolution image features, the skeletons contain more accurate information about the human pose, human motion, and basic human shape (e.g., body height) [24]. However, HPE utilizes joint coordinates to represent human pose, while HMR estimates joint rotations as SMPL pose parameters. Due to the difference in pose representation between these two tasks, previous video-based HMR methods do not effectively utilize the skeletons. PMCE [30] first introduces

HPE into video-based HMR by integrating the skeletons and image features. However, it treats the skeleton as a supplementary feature, ignoring the advantages of different structural information of the skeleton in pose, shape, and motion estimation. Moreover, PMCE directly predicts mesh coordinates without using the human prior, often leading to self-interactions, unreasonable poses and shapes.

Based on the above observations, we propose a novel semi-**A**nalytical **R**egressor using disen**T**angled **S**keletal Representations (**ARTS**) as shown in Figure 1 (b), which incorporates skeletons to alleviate the problems above through analytics and learning methods. We divide the human mesh recovery task into two parts: 1) 3D skeleton estimation and disentanglement and 2) regressing SMPL parameters from the disentangled skeletal representations through a semi-analytical regressor. Firstly, we propose a skeleton estimation and disentanglement module to estimate 3D skeletons from a video and decouple them into disentangled skeletal representations (i.e., joint position, bone length, and human motion). This disentanglement allows the following modules to focus on different information about human pose, shape, and motion. Secondly, to fully utilize the disentangled skeletal representations, we introduce a semi-analytical SMPL regressor consisting of three modules, which combines analytics and learning methods to estimate SMPL parameters. Specifically, a Temporal Inverse Kinematics (TIK) module is proposed to derive initial SMPL pose parameters from the joint position. The inverse kinematics improves the accuracy and robustness of SMPL pose estimation. To alleviate the challenge of ineffective shape fitting, we design a Bone-guided Shape Fitting (BSF) module, which combines analytics with MLP to regress the initial SMPL shape parameters from the bone length. Besides, we propose a Motion-Centric Refinement (MCR) module that employs cross-attention to obtain motion-centric features from image features and human motion representation. Then, the motion-centric features are utilized to refine the initial SMPL parameters and enhance the temporal consistency of human mesh. Our model is evaluated on popular 3D human mesh recovery benchmarks, outperforming previous state-of-the-art video-based HMR methods.

Our main contributions are summarized as follows:

- We propose a semi-Analytical Regressor using disenTangled Skeletal representations (ARTS) that combines analytics with learning methods, which effectively leverages structural information of skeletons to improve both per-frame accuracy and temporal consistency.
- In semi-analytical regressor, we carefully design three components, i.e., Temporal Inverse Kinematics (TIK), Bone-guided Shape Fitting (BSF), and Motion-Centric Refinement (MCR), for learning accurate and temporally consistent human pose, shape, and motion, respectively.
- Our method achieves state-of-the-art performance on multiple 3D human mesh recovery benchmarks. Particularly, in cross-dataset evaluation, ARTS reduces MPJPE and MPVPE by 10.7% and 10.5% on the 3DPW [21] dataset.

## 2 RELATED WORK

### 2.1 3D Human Pose Estimation

Benefiting from the advance of 2D pose detections [31, 32], many recent works for 3D HPE are based on the 2D-to-3D lifting pipeline

[1]. Some 3D pose estimation methods [33–36] employ Graph Convolutional Networks (GCNs) [37] to extract spatial-temporal information from the skeletons by treating the human skeleton as a graph. In recent years, Transformer-based methods [25–27, 29] have become popular, which carefully design serial or parallel, global or local Transformer [38] networks to explore the spatial and temporal dependencies of 2D skeletons. The achievements in 3D HPE demonstrate that the skeletons contain sufficient spatial and temporal information about human body, which can be leveraged to produce accurate and temporally consistent human mesh.

## 2.2 Image-based 3D Human Mesh Recovery

Due to the easy access to images, many human mesh recovery methods take a single image as input. The image-based methods can be divided into two paradigms. The first is parametric methods based on the human model (e.g., SMPL [8]). Due to the complexity of the SMPL estimation, some methods incorporate prior knowledge to assist in estimating SMPL parameters, such as body silhouette [39], semantic body part segmentation [40, 41], bounding box [4], and kinematic prior [19, 42, 43]. Moreover, the skeletons from 3D HPE can assist the human mesh regression. For example, Nie *et al.* [44] aim to learn a good pose representation that disentangles pose-dependent and view-dependent features from human skeleton data. However, the pose-dependent features capture only the overall pose and lack disentangled semantic details. PC-HMR [45] uses the estimated human pose to calibrate the human mesh estimated by the off-the-shelf HMR methods, serving as a time-consuming post-processing. The other paradigm is non-parametric methods, which directly estimate the coordinates of each mesh from an image. For instance, Pose2Mesh [3] designs a multi-stage MeshNet to upsample sparse 3D skeletons to human mesh. GTRS [5] introduces a graph Transformer network to reconstruct human mesh from a 2D human pose. Although these image-based methods have achieved remarkable performance in accuracy, they tend to produce unsmooth human motion when applied to videos.

## 2.3 Video-based 3D Human Mesh Recovery

Compared to image-based HMR methods, video-based HMR methods simultaneously recover accurate and temporally consistent human mesh. The previous video-based methods mainly focus on designing temporal extraction and fusion networks to enhance temporal consistency. For instance, VIBE [10], MEVA [11], and TCMR [12] carefully design Gated Recurrent Units (GRUs) based temporal extraction networks, which are utilized to smooth human motion but lack sufficient ability to model long-term dependencies. Consequently, most methods utilize Transformer-based temporal networks. MEAD [46] proposes a spatial and temporal Transformer to parallel model these two dependencies. GLoT [14] proposes a global and local Transformer to decompose the modeling of long-term and short-term temporal correlations. Bi-CF [15] introduces a bi-level Transformer to model temporal dependencies in a video clip and among different clips. UNSPAT [17] proposes a spatiotemporal Transformer to incorporate both spatial and temporal information without compromising spatial information. Despite the complicated design of these temporal networks, the insufficient spatial information and noises of image features unavoidably lead to limited

performance. Although skeleton prior is commonly used in image-based methods, it is often ignored by video-based methods. Sun *et al.* [47] also uses skeleton-disentangled representation, but its disentanglement refers to only extracting the skeletons from image features without HPE methods. Besides, it does not decompose different information within skeletons so it may not fully utilize the skeleton data. PMCE [30] is the first to incorporate 3D HPE methods in video-based HMR. However, it directly utilizes cross-attention to fuse image features and the skeletons, which ignores the structural information of the skeletons. Additionally, PMCE directly regresses mesh coordinates without incorporating sufficient human prior (e.g., SMPL), which often results in self-interactions, unreasonable body poses and shapes. In contrast, our ARTS decouples the skeletons into joint position, human motion, and bone length and exploits the advantages of the disentangled skeletal representations in regressing different parameters of the SMPL model. This makes our method achieve better per-frame accuracy and temporal consistency.

## 3 METHODOLOGY

### 3.1 Overall Framework

The overall framework of our semi-Analytical Regressor using disenTangled Skeletal representations for HMR (ARTS) is illustrated in Figure 2, which primarily consists of two parts: 1) 3D skeleton estimation and disentanglement and 2) regressing human mesh from the disentangled skeletal representations and image features through a semi-analytical SMPL regressor. In detail, given a video sequence $V = \{I_t\}_{t=1}^{T}$ with $T$ frames. A pre-trained ResNet50 [9] from SPIN [48] is utilized to extract image features $F \in \mathbb{R}^{T \times 2048}$ for each frame. For 3D skeleton estimation and disentanglement, we first employ the off-the-shelf 2D pose detector [31, 32] to estimate 2D skeletons $S^{2D} \in \mathbb{R}^{T \times K \times 2}$ and then design a dual-stream Transformer network to lift 2D skeletons to 3D skeletons $S^{3D} \in \mathbb{R}^{T \times K \times 3}$, where $K$ denotes the number of human keypoints. After that, we further decouple the 3D skeletons into disentangled skeletal representations (i.e., joint position $J$, human motion $M$, and bone length $B$). In the second part, the semi-analytical SMPL regressor regresses SMPL parameters from disentangled skeletal representations and image features. In regressor, we propose Temporal Inverse Kinematics (TIK) to regress initial SMPL pose parameters $\theta_{init} \in \mathbb{R}^{72}$ of mid-frame from joint position and image features. The Bone-guided Shape Fitting (BSF) module is introduced to predict bone-aligned SMPL shape parameters $\beta_{init} \in \mathbb{R}^{10}$ of mid-frame from bone length. Finally, we design a Motion-Centric Refinement (MCR) to fuse image features with human motion guidance and employ motion-centric features $F' \in \mathbb{R}^{2048}$ to align SMPL parameters with the image cues. We elaborate on each part in the following sections.

### 3.2 Skeleton Estimation and Disentanglement

*3.2.1 **3D Human Pose Estimation.*** We first utilize the off-the-shelf 2D pose detectors [31, 32] to obtain the 2D skeletons from the image sequence. Subsequently, we design a simple yet effective $L_1$ blocks dual-stream Transformer network to lift the 2D skeletons to 3D skeletons. Specifically, we first project the 2D skeletons $S^{2D} \in \mathbb{R}^{T \times K \times 2}$ into high-dimensional features $X \in \mathbb{R}^{T \times K \times C_1}$ and incorporate spatial and temporal positional embeddings. Then, we

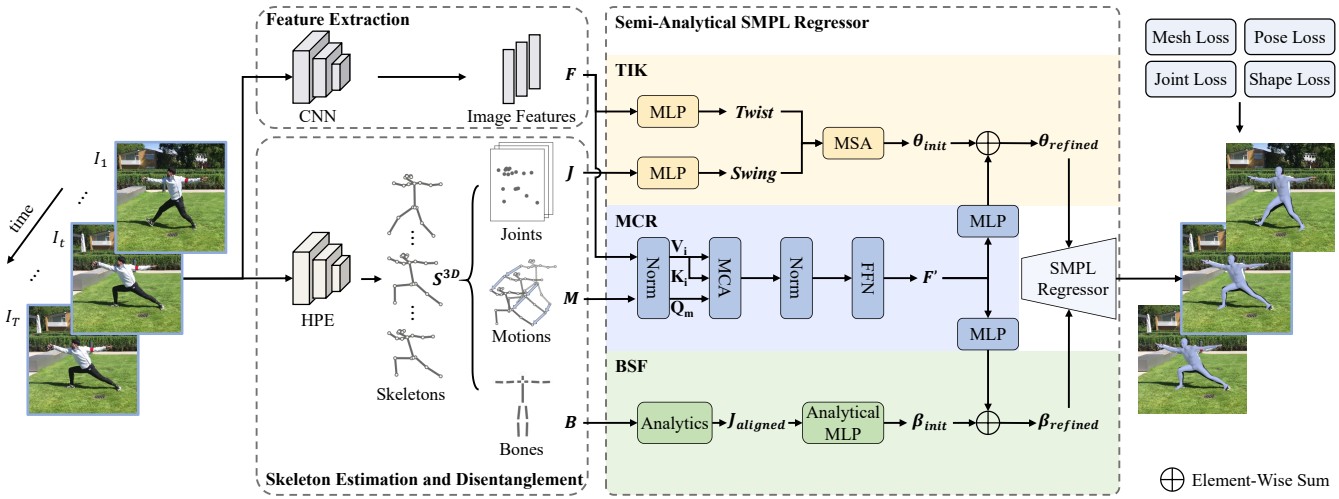

**Figure 2: Overview of the proposed ARTS. Given a video sequence, ResNet [9] is utilized to extract the image features $F$ of each frame. We estimate the 3D skeletons and decouple them into joints, motions, and bones. Then, in the semi-analytical SMPL regressor, Temporal Inverse Kinematics (TIK) obtains initial SMPL pose parameters $\theta_{init}$ from joints and image features. Bone-guided Shape Fitting (BSF) gets bone-aligned SMPL shape parameters $\beta_{init}$ from bones. Moreover, we utilize motions to guide the fusion of image features and use motion-centric features $F'$ to refine SMPL parameters. Finally, ARTS feeds the refined SMPL parameters $\theta_{refined}$, $\beta_{refined}$ to the SMPL regressor to generate the human mesh.**

feed $X$ to the dual-stream Transformer and the output features $X' \in \mathbb{R}^{T \times K \times C_1}$ are fused as the subsequent block's input, which can be expressed as:

$$X' = \text{MSA}_{ST}(X + PE_S + PE_T) + \text{MSA}_{TS}(X + PE_S + PE_T), \quad (1)$$

where $\text{MSA}_{ST}$ denotes spatial-temporal Transformer, $\text{MSA}_{TS}$ denotes temporal-spatial Transformer, $PE_S$ and $PE_T$ represents spatial and temporal embedding, respectively. Finally, the high-dimensional features $X'$ are mapped into the 3D skeletons $S^{3D} \in \mathbb{R}^{T \times K \times 3}$.

*3.2.2* ***Skeleton Disentanglement.*** Different from the previous video-based method [30] that treats the skeletons as supplementary features, we decouple the 3D skeletons $S^{3D}$ into the disentangled skeletal representations (i.e., joint position $J$, bone length $B$, and human motion $M$). Specifically, joint position $J \in \mathbb{R}^{T \times K \times 3}$ represents the positions of all joints, which contains accurate information about human pose. The bone length $B \in \mathbb{R}^N$ represents the temporally average length of each bone within the video sequence, where $N$ denotes the number of bones. This average bone length contains temporally consistent basic body shape information. For human motion, We calculate the detailed motion of each frame $M_t \in \mathbb{R}^{(T-1) \times K \times 3}$ and then concat temporally average human motion $M_0 \in \mathbb{R}^{1 \times K \times 3}$ to get the final human motion $M \in \mathbb{R}^{T \times K \times 3}$, which represents the overall and detailed human movements. The equations of human motion and bone length are as follows:

$$M_t = \{S_t^{3D} - S_{t-1}^{3D}\}_{t=2}^T, M_0 = avg(M_t), M = concat[M_0, M_t], \quad (2)$$

$$B_n = \frac{1}{T} \sum_{t=1}^T \|S_{t,i}^{3D} - S_{t,j}^{3D}\|, \quad (3)$$

where $t$ represents $t^{th}$ frame, $avg(\cdot)$ represents averaging over time, $n$ represents $n^{th}$ bone, $i$ and $j$ denote two joints of the $n^{th}$ bone.

### 3.3 Semi-Analytical SMPL Regressor

As shown in Figure 2, we propose a semi-analytical SMPL regressor to predict the SMPL parameters from the disentangled skeletal representations and image features. Semi-analytical regressor combines accurate analytical methods with flexible learning methods, which can generate accurate human mesh from skeletons and maintain robustness to the errors of skeleton estimation. Specifically, the joint position can provide a more accurate human pose. The bone length contains consistent basic body shape and human motion can provide accurate and smooth human movement. Based on these observations, we design three modules (i.e., TIK, BSF, and MCR) to exploit the advantages of the disentangled skeletal representations in regressing different parameters of the SMPL model.

*3.3.1* ***Temporal Inverse Kinematics.*** We regress accurate SMPL pose parameters from the joint position and image features. Following HybrIK [19], we first decompose the SMPL pose into swing among neighboring joints and twist that represents the angle along the bone direction. Swing can be accurately derived from joint position, while twist cannot be directly obtained from the coordinates of the joints, so we regress the twist of each joint from image features. This decomposition allows us to tackle the SMPL pose estimation by separately handling the joint-based swing calculation and image-based twist estimation, significantly reducing the complexity of the SMPL pose estimation. Different from HybrIK, which solely relies on analytical methods and ignores temporal cues, we utilize MLP and self-attention to obtain temporally consistent SMPL joint rotation from swing and twist. This process is expressed as follows:

$$swing = \text{MLP}(J), twist = \text{MLP}(F), \quad (4)$$

$$\theta_{init} = \text{MSA}(\text{MLP}(concat[swing, twist])), \quad (5)$$

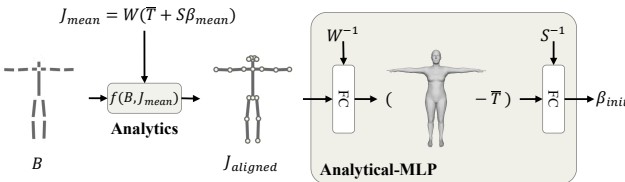

**Figure 3: Illustration of the bone-guided shape fitting. Analytics and Analytical-MLP are utilized to map bone length into the initial SMPL shape parameters.**

where $swing \in \mathbb{R}^{T \times K' \times 6}$ denotes 6D rotations [49] among joints, $twist \in \mathbb{R}^{T \times (K'-1) \times 2}$ denotes twist angles of each joints (except root joint), $K'$ denotes the number of SMPL keypoints, $\theta_{init} \in \mathbb{R}^{72}$ denotes temporally consistent SMPL pose parameters of mid-frame, $MSA(\cdot)$ represents multi-head self-attention.

*3.3.2* ***Bone-Guided Shape Fitting.*** As shown in Figure 3, we regress bone-aligned SMPL shape parameters from the bone length. According to the SMPL[8], the joint position in the rest pose is a function of shape parameters. Given the mean shape parameters $\beta_{mean} \in \mathbb{R}^{10}$ of SMPL template, the joint position is calculated as:

$$J_{mean} = W(\bar{T} + S\beta_{mean}), \tag{6}$$

where $\bar{T} \in \mathbb{R}^{6890 \times 3}$ is the SMPL mesh template, $S \in \mathbb{R}^{6890 \times 3 \times 10}$ is the shape blend matrix that maps shape parameters to the offsets of template, $W \in \mathbb{R}^{K \times 6890}$ is the joint regression matrix that obtains joints from human mesh. We can obtain rough shape parameters from the bone length since the bone length $B$ contains the body height and reflects body weight [3]. We first align the rest pose $J_{mean}$ with bone length to get the bone-aligned pose $J_{aligned} \in \mathbb{R}^{K \times 3}$ through an analytical transform $f(B, J_{mean})$ along the kinematic tree:

$$J_{aligned,j} = J_{mean,i} + B_{i,j} \frac{J_{mean,j} - J_{mean,i}}{\|J_{mean,j} - J_{mean,i}\|}, \tag{7}$$

where $B_{i,j}$ denotes the bone length between parent joint $i$ and child joint $j$. Then, we can derive the bone-aligned SMPL shape parameters $\beta'_{init} \in \mathbb{R}^{10}$ from the bone-aligned pose:

$$\beta'_{init} = S^{-1}(W^{-1}J_{aligned} - \bar{T}). \tag{8}$$

However, relying solely on the analytical method makes the shape estimation highly sensitive to errors and noises in bone length. Therefore, instead of directly using the analytical transform, we initialize the MLP with the pseudo-inverse matrix $S^{-1}$ and $W^{-1}$. Then, we train the analytical-MLP transform $g(J_{aligned}, \bar{T})$ to fine-tune the mapping from bone-aligned pose to shape parameters. This process can be expressed as follows:

$$\beta_{init} = FC_{S^{-1}}(FC_{W^{-1}}(J_{aligned}) - \bar{T}), \tag{9}$$

where $FC_{S^{-1}}(\cdot)$ and $FC_{W^{-1}}(\cdot)$ are the full-connected layer initialized by pseudo-inverse shape blend matrix and joint regression matrix.

*3.3.3* ***Motion-Centric Refinement.*** Due to the sparsity of skeleton information, the SMPL parameters regressed from joint position and bone length lack human details (e.g., accurate shapes), which still need to be refined from images. Therefore, we utilize human motion to guide the fusion of image features and then refine the initial SMPL parameters. In detail, we first project human motion $M$ to

high-dimension features as the queries $Q_m \in \mathbb{R}^{T \times C_2}$ and projects image features $F$ to the keys $K_i \in \mathbb{R}^{T \times C_2}$ and values $V_i \in \mathbb{R}^{T \times C_2}$. Then, we employ $L_2$ layers cross-attention $MCA(\cdot)$ and feedforward networks FFN to obtain the motion-centric features $F' \in \mathbb{R}^{2048}$, which can be expressed as follows:

$$F' = FFN(MCA(Q_m, K_i, V_i)). \tag{10}$$

This fusion manner makes the network focused on human movement. Moreover, we utilize MLP heads to refine the initial SMPL parameters from motion-centric features. Finally, we utilize the refined SMPL pose and shape parameters $\theta_{refined}, \beta_{refined}$ to generate the final human mesh.

### 3.4 Loss Function

The skeleton estimation and disentanglement module is trained with the 3D joint loss $\mathcal{L}_{joint}$ to supervise the 3D skeletons of all frames. Then, the whole network is supervised by four losses: mesh vertex loss $\mathcal{L}_{mesh}$, 3D joint loss $\mathcal{L}_{joint}$, SMPL pose loss $\mathcal{L}_{pose}$, SMPL shape loss $\mathcal{L}_{shape}$, The final loss is calculated as:

$$\mathcal{L} = \lambda_m \mathcal{L}_{mesh} + \lambda_j \mathcal{L}_{joint} + \lambda_p \mathcal{L}_{pose} + \lambda_s \mathcal{L}_{shape}, \tag{11}$$

where $\lambda_m = 1$, $\lambda_j = 1$, $\lambda_p = 0.06$, and $\lambda_s = 0.06$ in ARTS. The $\lambda$ parameters ensure that the values of various loss functions are maintained at the same range.

## 4 EXPRIMENTS

### 4.1 Datasets and Evaluation Metrics

*4.1.1* ***Datasets.*** Following previous methods [14, 15, 30], we train our model on the mixed 2D and 3D datasets. For 3D datasets, 3DPW [21], MPI-INF-3DHP [23], and Human3.6M [22] contain the annotations of 3D joints and SMPL parameters. For 2D datasets, COCO [50] and MPII [51] contain 2D joint annotation with pseudo-SMPL parameters from NeuralAnnot [52]. To compare with previous methods, we evaluate the performance of our model on the 3DPW, MPI-INF-3DHP, and Human3.6M datasets.

*4.1.2* ***Evaluation Metrics.*** To evaluate per-frame accuracy, we employ the mean per joint position error (MPJPE), Procrustes-aligned MPJPE (PA-MPJPE), and mean per vertex position error (MPVPE). These metrics measure the difference between the predicted mesh position and ground truth in millimeters ($mm$). To evaluate temporal consistency, we utilize the acceleration error (Accel) proposed in HMMR [53]. This metric calculates the average difference in acceleration of joints, which is measured in $mm/s^2$.

### 4.2 Implementation Details

Consistent with the previous methods [14, 15, 30], we set the input sequence length $T$ to 16 and utilize the pre-trained ResNet50 from SPIN [48] to extract image features of each frame. We train the network in two stages. Firstly, we train the 3D skeleton estimation network with all of the 3D and 2D datasets with a batch size of 64 and a learning rate of $1 \times 10^{-5}$ for 60 epochs. For 2D pose detectors in the skeleton estimation network, we adopt CPN [32] for Human3.6M and ViTPose [31] for 3DPW and MPI-INF-3DHP. In the second stage, we load the weights of the 3D skeleton estimation network and train the whole model with only 3D datasets. We train the whole network for 30 epochs with a batch size of 32 and a

**Table 1: Evaluation of state-of-the-art methods on 3DPW, MPI-INF-3DHP, and Human3.6M datasets. All methods use pre-trained ResNet-50 [46] as the backbone to extract image features except MAED [46]. Bold: best; Underline: second bset.**

| Method | 3DPW | | | | MPI-INF-3DHP | | | Human3.6M | | |
|---|---|---|---|---|---|---|---|---|---|---|
| | MPJPE ↓ | PA-MPJPE ↓ | MPVPE ↓ | Accel ↓ | MPJPE ↓ | PA-MPJPE ↓ | Accel ↓ | MPJPE ↓ | PA-MPJPE ↓ | Accel ↓ |
| VIBE (CVPR'20) [10] | 91.9 | 57.6 | - | 25.4 | 103.9 | 68.9 | 27.3 | 78.0 | 53.3 | 27.3 |
| TCMR (CVPR'21) [12] | 86.5 | 52.7 | 102.9 | 6.8 | 97.3 | 63.5 | 8.5 | 73.6 | 52.0 | 3.9 |
| MEAD (ICCV'21) [46] | 79.1 | 45.7 | 92.6 | 17.6 | 83.6 | 56.2 | - | 56.4 | 38.7 | - |
| MPS-Net (CVPR'22) [13] | 84.3 | 52.1 | 99.7 | 7.6 | 96.7 | 62.8 | 9.6 | 69.4 | 47.4 | 3.6 |
| Zhang (CVPR'23) [54] | 83.4 | 51.7 | 98.9 | 7.2 | 98.2 | 62.5 | 8.6 | 73.2 | 51.0 | 3.6 |
| GLoT (CVPR'23) [14] | 80.7 | 50.6 | 96.3 | 6.6 | 93.9 | 61.5 | 7.9 | 67.0 | 46.3 | 3.6 |
| Bi-CF (MM'23) [15] | 73.4 | 51.9 | 89.8 | 8.8 | 95.5 | 62.7 | 7.7 | 63.9 | 46.1 | **3.1** |
| PMCE (ICCV'23) [30] | 69.5 | 46.7 | 84.8 | **6.5** | 79.7 | 54.5 | **7.1** | 53.5 | 37.7 | **3.1** |
| UNSPAT (WACV'24) [17] | 75.0 | **45.5** | 90.2 | 7.1 | 94.4 | 60.4 | 9.2 | 58.3 | 41.3 | 3.8 |
| ARTS (Ours) | **67.7** | 46.5 | **81.4** | **6.5** | **71.8** | **53.0** | 7.4 | **51.6** | **36.6** | **3.1** |

learning rate of $3 \times 10^{-5}$. We set layer number $L_1 = 3$ and $L_2 = 1$ and use feature dimension $C_1 = 256$ and $C_2 = 512$. The experiments are implemented by PyTorch on a single NVIDIA RTX 4090 GPU.

## 4.3 Comparison with State-of-the-art Methods

*4.3.1 Comparison with Video-based Methods.* As shown in Table 1, we report the results of our model on popular HMR benchmarks: 3DPW, MPI-INF-3DHP, and Human3.6M. Our ARTS surpasses existing state-of-the-art video-based methods in both per-frame accuracy and temporal consistency metrics. Specifically, compared to the previous state-of-the-art method PMCE [30], our model achieves a reduction of 2.6% (from 69.5$mm$ to 67.7$mm$), 9.9% (from 79.7$mm$ to 71.8$mm$), and 3.6% (from 53.5$mm$ to 51.6$mm$) in MPJPE metric on 3DPW, MPI-INF-3DHP, and Human3.6M datasets, respectively. Next, we analyze the limitations of previous video-based methods. GLoT [14], Bi-CF [15], and UNSPAT [17] regress SMPL parameters based on the low-resolution image features. Despite the design of complex networks to extract spatial-temporal features, the image features extracted by the backbone lack sufficient spatial information about the human body and contain various noises, resulting in inaccurate and unsmooth human mesh. Although UN-SPAT [17] and MEAD [46] have slightly lower PA-MPJPE than ours on the 3DPW dataset, MEAD utilizes ViT [55] as the backbone and their performance on other error metrics and datasets is much higher. PMCE [30] also incorporates 3D skeletons, but it integrates skeletons as a low-dimension feature. This limitation results in lower reconstruction accuracy of PMCE. Although PMCE achieves a marginally better Accel than ours by 0.3$mm/s^2$ on the MPI-INF-3DHP dataset, it utilizes more 2D datasets with diverse scenes for the training of the SMPL regression network and its estimation error MPJPE is significantly higher than ours by 7.9$mm$.

Different from previous methods, our ARTS effectively utilizes the disentangled skeletal representations for different SMPL parameters regression, including accurate joint position (lower MPJPE and PA-MPJPE), temporally consistent bone length (lower MPVPE),

**Table 2: Evaluation of state-of-the-art methods on cross-domain generalization. All methods do not use 3DPW for training but evaluate on 3DPW dataset.**

| | Method | 3DPW | | | |
|---|---|---|---|---|---|
| | | MPJPE ↓ | PA-MPJPE ↓ | MPVPE ↓ | Accel ↓ |
| Image-based | Pose2Mesh [3] (ECCV'20) | 88.9 | 58.3 | 106.3 | 22.6 |
| | HybrIK [19] (CVPR'21) | 80.0 | **48.8** | 94.5 | 25.1 |
| | GTRS [5] (MM'22) | 88.5 | 58.9 | 106.2 | 25.0 |
| | CLIFF [4] (CVPR'22) | 85.4 | 53.6 | 100.5 | - |
| | SimHMR [6] (MM'23) | 81.3 | 49.5 | 102.8 | - |
| | ScoreHMR [7] (CVPR'24) | - | 50.5 | - | 11.1 |
| Video-based | VIBE [10] (CVPR'20) | 93.5 | 56.5 | 113.4 | 27.1 |
| | TCMR [12] (CVPR'21) | 95.0 | 55.8 | 111.5 | 7.0 |
| | MPS-Net [13] (CVPR'22) | 91.6 | 54.0 | 109.6 | 7.5 |
| | INT [16] (ICLR'23) | 90.0 | 49.7 | 105.1 | 23.5 |
| | GLoT [14] (CVPR'23) | 89.9 | 53.5 | 107.8 | 6.7 |
| | PMCE [30] (ICCV'23) | 81.6 | 52.3 | 99.5 | 6.8 |
| | Bi-CF [15] (MM'23) | 78.3 | 53.7 | 95.6 | 8.6 |
| | ARTS (Ours) | **69.9** | **48.8** | **85.6** | **6.6** |

and precise human motion (lower Accel). Therefore, we achieve more accurate and temporally consistent human mesh recovery.

*4.3.2 Cross-dataset Evaluation Results.* To evaluate the generalization ability of our model, we conduct cross-dataset evaluation. Following previous methods [14, 15, 30], we train our model on the MPI-INF-3DHP and Human3.6M datasets and evaluate it on 3DPW dataset. As illustrated in Table 2, compared with image-based and video-based methods that report cross-dataset evaluation results, our ARTS shows a significant improvement in both per-frame accuracy and temporal consistency. Specifically, compared to Bi-CF [15],

**Table 3: Ablation study for different components in semi-analytical SMPL regressor on 3DPW dataset.**

| Module | | | 3DPW | | | |
|---|---|---|---|---|---|---|
| TIK | MCR | BSF | MPJPE ↓ | PA-MPJPE ↓ | MPVPE ↓ | Accel ↓ |
| ✗ | ✗ | ✗ | 72.2 | 51.5 | 87.8 | 15.5 |
| ✓ | ✗ | ✗ | 69.5 | 47.6 | 85.3 | 10.6 |
| ✗ | ✓ | ✗ | 70.5 | 48.9 | 86.1 | 6.7 |
| ✗ | ✗ | ✓ | 70.7 | 48.3 | 83.6 | 15.2 |
| ✓ | ✗ | ✓ | 68.9 | 46.6 | 82.3 | 11.4 |
| ✓ | ✓ | ✗ | 68.7 | 46.9 | 83.4 | 6.6 |
| ✗ | ✓ | ✓ | 68.6 | 47.1 | 82.6 | 6.7 |
| ✓ | ✓ | ✓ | **67.7** | **46.5** | **81.4** | **6.5** |

our model brings obvious improvements in the accuracy metrics MPJPE, PA-MPJPE, and MPVPE by 10.7% (from 78.3$mm$ to 69.9$mm$), 8.9% (from 53.7$mm$ to 48.9$mm$), and 10.5% (from 95.6$mm$ to 85.6$mm$), respectively. Additionally, we obtain a further 23.3% reduction (from 8.6$mm/s^2$ to 6.6$mm/s^2$) in the temporal consistency metric Accel. These results demonstrate the robust cross-domain generalization ability of our model, which is primarily attributed to our effective utilization of disentangled skeletal representations. The image features extracted by the CNN backbone [9] vary significantly across different datasets, while the representation of the skeleton is quite similar across datasets. Therefore, the full utilization of the skeleton enhances the overall robustness of our model.

## 4.4 Ablation Study

*4.4.1 Component-wise Ablation of Semi-analytical SMPL Regressor.* We conduct experiments on the 3DPW dataset to illustrate the effects of three proposed components in semi-analytical SMPL regressor: Temporal Inverse Kinematics (TIK), Bone-guided Shape Fitting (BSF), and Motion-Centric Refinement (MCR). As shown in Table 3, the absence of the entire semi-analytical SMPL regressor (Row 1), which does not utilize skeletons, leads to an increase in MPJPE and Accel by 4.5$mm$ and 9.0$mm/s^2$, respectively. Regarding each module, the utilization of TIK (Row 2) results in a decrease of 2.7$mm$ in MPJPE, while the MCR (Row 3) module notably decreases Accel by 8.8$mm/s^2$. Furthermore, the inclusion of BSF (Row 4) leads to a reduction of 4.2$mm$ in MPVPE. Removing these modules from the overall model yields similar effects (Row 5-8), demonstrating the significant role of TIK in enhancing pose regression accuracy, BSF in improving shape regression accuracy, and MCR in enhancing the smoothness of human motion. These results demonstrate the critical importance of each component in achieving superior performance.

*4.4.2 Different Inverse Kinematics.* We conduct experiments on the 3DPW dataset to explore the effectiveness of our proposed Temporal Inverse Kinematics (TIK). Compared to Frame-based Inverse Kinematics (FIK) similar to HybrIK [19], TIK utilizes multi-head self-attention to fuse temporal cues of joint rotations. We compared the robustness of TIK and FIK when adding Gaussian noise to the skeleton. The results are shown in Table 4 (Row 1-6),

**Table 4: Ablation study for different designs of the inverse kinematics, shape fitting, and feature fusion on 3DPW.**

| Module | | 3DPW | | | |
|---|---|---|---|---|---|
| Aspect | Method | MPJPE ↓ | PA-MPJPE ↓ | MPVPE ↓ | Accel ↓ |
| Frame-IK | 0% | **68.3** | **46.8** | **82.1** | **9.4** |
| | 2% | 71.1 | 48.9 | 85.4 | 19.3 |
| | 10% | 97.9 | 67.2 | 116.0 | 57.2 |
| Temporal-IK | 0% | **67.7** | **46.5** | **81.4** | **6.5** |
| | 2% | 68.3 | 46.8 | 82.1 | 7.2 |
| | 10% | 76.1 | 52.5 | 91.4 | 26.9 |
| Shape Fitting | Analytics | 69.5 | 48.0 | 83.2 | **6.5** |
| | MLP | 68.4 | 46.9 | 82.4 | 6.6 |
| | Analytical-MLP | **67.7** | **46.5** | **81.4** | **6.5** |
| Feature Fusion | GRU | 68.8 | 47.1 | 82.9 | 7.6 |
| | SA | 68.9 | 47.4 | 82.4 | 7.7 |
| | Motion-CA | **67.7** | **46.5** | **81.4** | **6.5** |

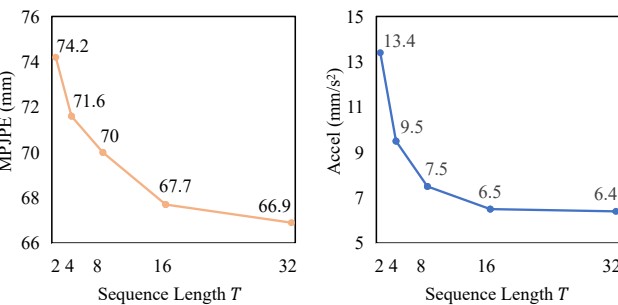

**Figure 4: Ablation study for different sequence lengths $T$ in terms of MPJPE (left) and Accel (right) on 3DPW dataset.**

when only 2% Gaussian noise is added, TIK can maintain the accuracy and consistency, while the Accel of FIK increases by 9.9$mm/s^2$ and its accuracy also decreases. When 10% noise is added, although both TIK and FIK have great errors, TIK exhibits a smaller decline (12.4%) than FIK (43.3%) in MPJPE. These results demonstrate that our TIK is more robust for video-based HMR.

*4.4.3 Different Shape Fitting Strategies.* As illustrated in Table 4 (Row 7-9), we investigate the impact of different shape fitting strategies. The results demonstrate that utilizing only the analytics leads to a reduction of 1.8$mm$ in MPVPE, whereas employing only the MLP results in a similar decrease of 1.0$mm$. Our Analytical-MLP achieves the best performance. The analytics-only strategy discussed in Section 3.3.2 and Equation 8 provides a shape with minimal error since the estimated bone length contains errors. The MLP-only strategy often leads to overfitting and ineffective body shape estimation due to insufficient body shape data. Our proposed analytical-MLP initializes MLP with the inverse matrix of joint regression and shape blend matrix, which offers strong prior for MLP while maintaining flexibility to errors of bone length.

*4.4.4 Different Feature Fusion Strategies.* Previous video-based HMR methods mainly focus on the design of image feature fusion.

**Table 5: Ablation study for different 2D pose detections on Human3.6M dataset.**

| Method | Human3.6M | | | |
|---|---|---|---|---|
| | MPJPE ↓ | PA-MPJPE ↓ | MPVPE ↓ | Accel ↓ |
| PMCE (SH [56]) | 56.4 | 39.0 | 64.5 | **3.2** |
| Ours (SH [56]) | **54.6** | **38.5** | **63.3** | 3.3 |
| PMCE (Detectron [57]) | 55.9 | 39.0 | 64.1 | **3.2** |
| Ours (Detectron [57]) | **53.7** | **38.1** | **62.8** | **3.2** |
| PMCE (CPN [32]) | 53.5 | 37.7 | 61.3 | **3.1** |
| Ours (CPN [32]) | **51.6** | **36.6** | **60.2** | **3.1** |
| PMCE (GT) | 36.3 | 26.8 | 46.2 | 2.2 |
| Ours (GT) | **33.7** | **24.8** | **45.7** | **2.0** |

As illustrated in Table 4 (Row 10-12), utilizing GRU networks similar to VIBE [10] and TCMR [12] leads to a reduction in Accel by $1.1mm/s^2$, whereas employing Self-Attention methods (SA) similar to GLoT [14] and Bi-CF [15] results in a reduction in Accel by $1.2mm/s^2$. Furthermore, both approaches exhibit a decrease in accuracy. ARTS leverages Motion-centric Cross-Attention (Motion-CA) to guide the fusion of image features, which enables the network to estimate temporally consistent human movements.

*4.4.5 **Impact of Sequence Lengths.*** For video-based HMR methods, sequence length has a direct impact on the performance. Although the 16-frames input length is commonly used for fair comparison, exploring the impact of different input lengths is also important. As shown in Figure 4, increasing the sequence length $T$ can improve the performance of both MPJPE (left) and Accel (right) on the 3DPW dataset. These results validate the effectiveness of our model in extracting and leveraging temporal cues of the human skeletons and image features within the sequence.

*4.4.6 **Impact of 2D Pose Detections.*** To investigate the impact of the 2D pose detectors used in the skeleton estimation and disentanglement, we conduct experiments using different 2D poses (e.g., SH [56], Detectron [57], CPN [32]) and Ground Truth (GT) pose as inputs. Table 5 compares PMCE[30] with ours on the Human3.6M dataset. The results indicate that ARTS outperforms PMCE across different 2D pose inputs. Moreover, our model has the potential to benefit from the development of 2D pose detectors and the experiment with GT inputs shows the lower bound of our method.

## 4.5 Qualitative Evaluation

Figure 5 shows the qualitative comparison among the previous state-of-the-art methods GLoT [14], PMCE [30], and our ARTS on the in-the-wild 3DPW dataset. Compared to GLoT that solely relies on image features, and PMCE that does not utilize the SMPL model as prior, our ARTS achieves more accurate and reasonable human meshes by using robust skeletons in various challenging scenarios, including self-interactions (Row 1), self-occlusion (Row 2), occlusion by others (Row 3), severe object occlusion (Row 4), low-light scene (Row 5), and image truncation (Row 6).

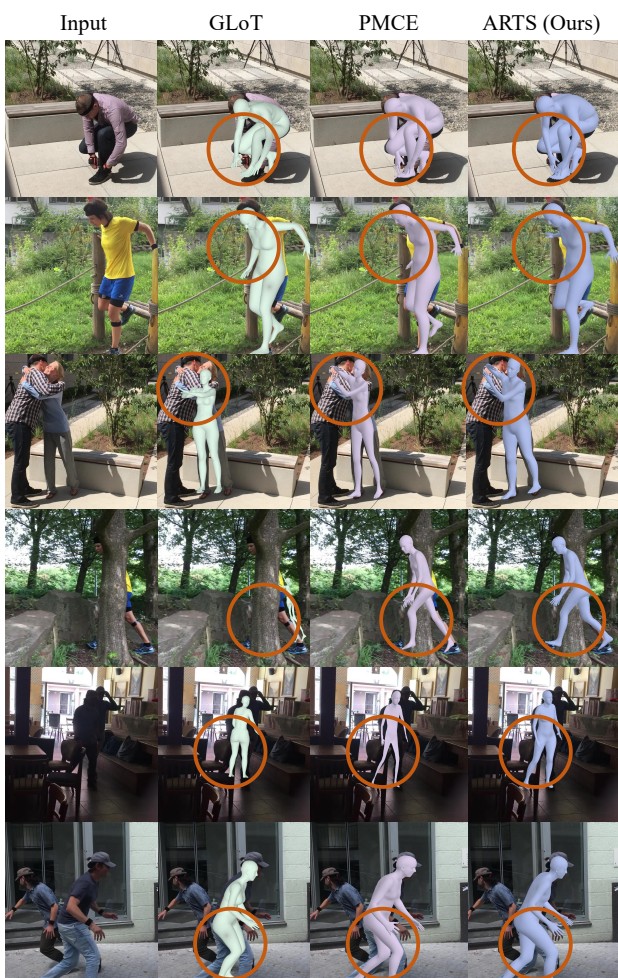

Input    GLoT    PMCE    ARTS (Ours)

**Figure 5: Qualitative comparison among GLoT (green mesh), PMCE (pink mesh) and our ARTS (blue mesh) on the challenging 3DPW dataset.**

## 5 CONCLUSION

In this paper, we present a novel semi-Analytical Regressor using disenTangled Skeletal representations for human mesh recovery from videos (ARTS), which mainly consists of two parts: 1) 3D skeleton estimation and disentanglement and 2) regressing SMPL parameters based on a semi-analytical SMPL regressor using the disentangled skeletal representations and image features. ARTS first estimates the 3D human skeletons from images and decouples them into joint position, human motion, and bone length. To estimate SMPL parameters from disentangled skeletal representations, we propose a semi-analytical SMPL regressor, which contains temporal inverse kinematics, bone-guided shape fitting, and motion-centric refinement modules. ARTS achieves state-of-the-art performance on multiple 3D human pose and shape estimation benchmarks. The cross-dataset evaluation and qualitative evaluation demonstrate the generalization ability and robustness of our ARTS. We hope our ARTS can further bridge the gap between human pose estimation and video-based human mesh recovery.

# ACKNOWLEDGMENTS

This work is supported by the National Natural Science Foundation of China (No.62373009).

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
