# OpenReview forum: "ARTS: Semi-Analytical Regressor using Disentangled Skeletal Representations for Human Mesh Recovery from Videos"
_acmmm.org/ACMMM/2024/Conference — MM2024 Poster_

### Official Review · Reviewer_7oj8 · 2024-05-26

**Rating:** 2
**Confidence:** 4

**Summary:**

This paper describes a method for 3D human pose and shape (HPS) estimation from a monocular video input. The key idea is to disentangle the human pose and shape information embedded in 2D and 3D human skeleton to guide the HPS estimation process. The proposed method is evaluated on common 3D human pose benchmarks, 3DPW, MPI-INF-3DHP, and Human3.6M. The results seem promising.

**Strengths:**

1. Promising results. The quantitative results on common benchmarks are quite promising. The qualitative video results look good.
2. The paper is easy to understand.
3. The idea of integrating skeleton sequence to refine temporal feature seems reasonable.

**Limitations:**

1. Novelty & Missing important reference. The key idea of using disentangled information of skeleton to guide the HPS estimation is quite similar to [1]. But the paper didn't mention this. Please cite and clearly tell the difference, most importantly the technical novelty. Except for the different architecture, what is the key innovation from a research view?
[1] Human Mesh Recovery From Monocular Images via a Skeleton-Disentangled Representation. ICCV, 2019.

2.Unfair comparisons. During evaluation, the image feature extraction part and 3D pose estimation part are only trained with similar 3D pose dataset, which seems to be fair. But the 2D pose estimation part is trained using an additional dataset, which makes the comparison less fair.

**Suitability:**

3

---

### Official Review · Reviewer_doQU · 2024-05-28

**Rating:** 4
**Confidence:** 4

**Summary:**

The paper is focused on the task of 3D human mesh recovery from videos. Specifically, the main idea is to estimate the 3D human mesh from videos with the help of the estimated 3D skeletons. Therefore, the proposed method leverages disentangled skeletal representations to effectively utilize the information contained in the skeletons, which are more accurate and robust than image features in representing human pose and motion. The method consists of two main parts: (1) the 3D skeleton estimation and disentanglement module, and (2) the semi-analytical SMPL regressor from the disentangled skeletal representations. Evaluations on three widely-used datasets show the effectiveness of the proposed method.

**Strengths:**

1. Novelty: The idea of the semi-analytical regressor using disentangled skeletal representations to estimate SMPL parameters is novel compared to existing methods that directly predict SMPL parameters from image features.
2. Technical correctness: The proposed method is technically sound and outperforms existing state-of-the-art methods, demonstrating its effectiveness.
3. Adequate evaluation: The paper provides a comprehensive evaluation of the proposed method using both per-frame accuracy and temporal consistency.
4. Clarity: The paper is well-written and easy to understand. The figures and video demos are clearly presented.
5. Applications: The proposed method has potential applications in various fields such as virtual reality, animation, gaming, and robotics.

**Limitations:**

1. Compared with the existing end-to-end pipeline, the proposed framework is complex with two CNN-based backbone networks and a regressor head. I suggest the authors compare the number of parameters among the proposed method and existing ones.
2. The performance of the proposed method relies on the performance of 2D pose estimation and 3D lifting. These two techniques may limit the applications of the proposed method.
3. The experiments are insufficient for the following aspects: 1) only ResNet-50 is used as the backbone, so it is better to use other backbones like the ViT-based ones to further improve the performance, 2) from Table 1 and Table 2, it seems that the temporal consistency of the ARTS is just competitive to the SOTA methods like GLoT and PMCE.
4. Basically, this paper considers a computer vision task using unimodal data, i.e., video, so it is moderately suitable for the MM conference.

**Suitability:**

2

---

### Official Review · Reviewer_s5nz · 2024-05-28

**Rating:** 4
**Confidence:** 3

**Summary:**

The paper introduces ARTS, a new method for 3D human mesh recovery from videos that uses disentangled skeletal representations. It includes a novel semi-analytical regressor with three modules: Temporal Inverse Kinematics, Bone-guided Shape Fitting, and Motion-Centric Refinement. The method demonstrates improved accuracy and temporal consistency over existing methods on benchmarks such as 3DPW, MPI-INF-3DHP, and Human3.6M.

**Strengths:**

1. The paper introduces a semi-analytical Regressor using disentangled Skeletal representations (ARTS) that combines analytical and learning methods to enhance 3D human mesh recovery accuracy and consistency.

2. The method design is generally technically sound to me.

3. The author provides extensive experimental results.

4. The result of the proposed method exceeds SOTA.

**Limitations:**

I have only one concern about the design of the method. It seems that previous works have similar disentangled designs. Could you explain some non-trivial differences between the proposed methods with the previous methods, such as [1,2]?

[1] Nie, Qiang, Ziwei Liu, and Yunhui Liu. "Unsupervised 3d human pose representation with viewpoint and pose disentanglement." Computer Vision–ECCV 2020: 16th European Conference, Glasgow, UK, August 23–28, 2020, Proceedings, Part XIX 16. Springer International Publishing, 2020.

[2] Luan, Tianyu, et al. "Pc-hmr: Pose calibration for 3d human mesh recovery from 2d images/videos." Proceedings of the AAAI Conference on Artificial Intelligence. Vol. 35. No. 3. 2021.

**Suitability:**

3

---

### Official Review · Reviewer_w3cj · 2024-05-28

**Rating:** 4
**Confidence:** 1

**Summary:**

The paper presents ARTS, a semi-Analytical Regressor using disentangled Skeletal representations for human mesh recovery from videos. The key innovation of ARTS lies in its approach to estimating 3D human meshes by leveraging disentangled information from skeletal data, which is more accurate and robust compared to traditional methods that rely on low-resolution image features. The method is composed of two main components: a skeleton estimation and disentanglement module, and a semi-analytical regressor that includes Temporal Inverse Kinematics (TIK), Bone-guided Shape Fitting (BSF), and Motion-Centric Refinement (MCR). The authors claim good performance on benchmarks such as 3DPW, MPI-INF-3DHP, and Human3.6M.

**Strengths:**

1) The semi-analytical regressor, which combines elements of traditional analytical methods with machine learning techniques, is an interesting hybrid approach that could potentially offer the benefits of other methodologies.
2) The paper provides a clear explanation of the limitations of current video-based human mesh recovery methods and presents a logical argument for why their approach addresses these issues.
3) The extensive experiments and comparisons to state-of-the-art methods demonstrate the effectiveness of ARTS.

**Limitations:**

1) While the paper claims state-of-the-art results, it is not clear how the performance of ARTS compares to the latest methods in terms of computational efficiency and scalability, which are critical for practical applications.
2) The paper could benefit from a more detailed discussion on the choice of network architecture and hyperparameters for the various modules, as these decisions can significantly impact the results.
3) The paper does not discuss potential failure cases or scenarios where ARTS might not perform as expected, which would be valuable for understanding the limitations and robustness of the approach.

**Suitability:**

2

---

### Meta-Review · Area_Chair_MaJW · 2024-06-29

**Recommendation:** Accept (Poster)
**Confidence:** 5

**Metareview:**

This paper presents ARTS, a semi-analytical Regressor using disentangled Skeletal representations for human mesh recovery from videos. Two reviewers agree to accept the paper with Weak Accept and Borderline. One reviewer changed the Borderline Accept to Borderline Reject because the technical novelty is incremental and the suitability of multimedia, and one reviewer gave a weak reject but did not read the author's rebuttal and give the final rating. All the reviewers agree that the proposed method is effective. It also offers a novel approach for effectively integrating multimodal data (i.e., skeletons and images) to improve the HMR. Therefore, the paper can be accepted.